# Roles of Neuropeptide S in Anesthesia, Analgesia, and Sleep

**DOI:** 10.3390/ph14050483

**Published:** 2021-05-19

**Authors:** Tetsuya Kushikata, Kazuyoshi Hirota, Junichi Saito, Daiki Takekawa

**Affiliations:** 1Department of Anesthesiology, Graduate School of Medicine, Hirosaki University, Zaifu 5, Hirosaki 0368562, Japan; hirotak@hirosaki-u.ac.jp (K.H.); saitoj@hirosaki-u.ac.jp (J.S.); 2Department of Anesthesia, Hirosaki University Hospital, Honcho 53, Hirosaki 0368563, Japan; takekawa@hirosaki-u.ac.jp

**Keywords:** Neuropeptide S, Neuropeptide S receptor (NPSR1), anesthesia, analgesia, sleep

## Abstract

Neuropeptide S (NPS) is an endogenous peptide that regulates various physiological functions, such as immune functions, anxiety-like behaviors, learning and memory, the sleep–wake rhythm, ingestion, energy balance, and drug addiction. These processes include the NPS receptor (NPSR1). The NPS–NPSR1 system is also significantly associated with the onset of disease, as well as these physiologic functions. For example, NPS is involved in bronchial asthma, anxiety and awakening disorders, and rheumatoid arthritis. In this review, among the various functions, we focus on the role of NPS in anesthesia-induced loss of consciousness; analgesia, mainly by anesthesia; and sleep–wakefulness. Progress in the field regarding the functions of endogenous peptides in the brain, including NPS, suggests that these three domains share common mechanisms. Further NPS research will help to elucidate in detail how these three domains interact with each other in their functions, and may contribute to improving the quality of medical care.

## 1. Introduction

Neuropeptide S (NPS) is an endogenous peptide. NPS-expressing cells are located in the vicinity of the locus coeruleus [1]. NPS receptor 1 (NPSR1) is a G protein-coupled receptor (GPCR) that is expressed primarily in the bronchus [2], several brain regions—including the cerebral cortex, thalamic nuclei, hypothalamus, subiculum, and amygdala [3]—and immune cells [4]. The NPSR1 has two main isoforms, NPSR1-A and NPSR1-B. The NPSR1-A encodes the shorter isoform with a 29-residue long distinct C-terminus, whereas the NPSR1-B has an alternate 3′ exon (E9b), encoding a larger protein with a distinct 35-residue C-terminus. NPSR1-A has more potent signal transduction [5]. These NPS receptors regulate several physiological processes, including immune functions [6], anxiety-like behaviors [7], learning and memory [8], the sleep–wake rhythm [9,10], ingestion [11], and energy balance [4]. The NPS–NPSR1 system is also associated with pathological conditions as well as these physiologic functions. For example, a single-nucleotide polymorphism of NPSR1 is significantly associated with disease, such as asthma [12], anxiety [13,14,15] and panic disorder [16], and rheumatoid arthritis [17]. NPS enhanced lymphocyte proliferation and promoted macrophage phagocytosis. These effects were accompanied by upregulated expression of pro-inflammatory cytokines. In the case of asthma, NPSR1 expression in eosinophils was observed [18]. In this review, among these various functions of NPS, we focused on the role of NPS in anesthesia-induced loss of consciousness; analgesia, mainly by anesthesia; and sleep–wakefulness.

## 2. NPS in Anesthesia

### 2.1. Anesthesia Involves Endogenous Sleep Circuit

Over 150 years have passed since the discovery of anesthesia; however, the mechanism of the loss of consciousness by anesthesia is still unknown [19]. General anesthesia is thought to consist of four elements: sedation, analgesia, inhibition of the adverse autonomic reflex, and muscle relaxation. The first clinically applied inhalation anesthesia, ether, satisfied these four elements by itself. Subsequently, various general anesthetics were developed. The anesthetics used in modern anesthesia practice include intravenous anesthetics, volatile anesthetics, and gaseous anesthetics. Modern anesthetics is a generic name for the various heterogeneous drugs that meet the four above-mentioned conditions. The pharmacological properties of general anesthetics vary depending on the type of anesthetic; however, several studies indicate that some anesthetics may share their mechanism of loss of consciousness with endogenous sleep. Gamma-aminobutyric acid (GABA)-ergic anesthetics [20] and alpha-adrenergic anesthetics [21] affect the activity of the endogenous sleep circuit in loss of consciousness. Based on this report, the activation of the endogenous sleep circuit would be involved in the mechanism of loss of consciousness by general anesthetics. Indeed, several studies suggested that there could be a common mechanism underlying the loss of consciousness induced by sleep and anesthesia [22,23,24]. Based on this idea, the endogenous sleep circuit is partially responsible for the loss of consciousness induced by anesthesia [25]. However, unconsciousness is only one element of general anesthesia. Therefore, there is no evidence that the mechanism of sleep is the same as the mechanism of general anesthesia [26]. We address, referring to revealed evidence, that sleep and general anesthesia have a shared mechanism for the loss of consciousness, but not for other properties of anesthesia such as inhibition of the autonomic reflex or relaxation of muscles.

Recently, new ideas have emerged in the field of the mechanism of anesthesia. These include manipulating neural circuits using techniques such as electrical stimulation, local pharmacology, optogenetics, and chemogenetics. In applying these techniques to anesthesia and sleep, we could make clearer the relationship between them. Local pharmacology involves intra-cerebroventricular injection (icv) of NPS. Such injections inhibit the neuronal activity of the sleep-induction-related ventral preoptic area and promote emergence. As this region is thought to be actively involved in the induction of loss of consciousness by general anesthetics, a study on the mechanism of anesthesia in this region using NPS could provide further evidence that sleep and anesthesia share a common mechanism regarding loss of consciousness.

### 2.2. Anesthesia Is a Heterogenic Phenomenon

The anesthetics used in current anesthesia management have various properties, as mentioned above. For example, barbiturate and propofol are ligands that bind to the GABA receptor [27]. On the other hand, ketamine and nitrous oxide act as antagonists of the *N*-methyl-D-aspartic acid (NMDA) receptor [28]. Dexmedetomidine is an alpha-2 agonist [29]. Therefore, the sedative mechanism of the various anesthetics is not uniform. Orexin (OX), an endogenous wake-promoting substance similar to NPS, inhibits the anesthetic duration of barbiturate [30], ketamine [31], isoflurane [32], sevoflurane [33], and propofol [34]. Conversely, an antagonist of the OX receptor prolonged the anesthetic duration of barbiturate [30] and ketamine [31]. These OX study results suggest that an endogenous sleep circuit would be included in the mechanism of loss of consciousness in various types of anesthetics. 

### 2.3. NPS Modulates General Anesthesia

We studied the effect of central NPS on the duration of general anesthesia by icv injection of NPS and the NPS receptor antagonist [D-Cys(tBu)(5)]NPS. Neuropeptide S or [D-Cys(tBu)(5)]NPS was administered by icv to rats, and the anesthetic duration of ketamine and propofol was measured. Neuropeptide S (from 1 nmol to 30 nmol) decreased the ketamine and propofol anesthesia duration. The reaction curve was bell-shaped. 20 nmol of [D-Cys(tBu)(5)]NPS icv antagonized the effect of NPS 1 nmol on the anesthetic duration of ketamine and propofol. The icv administration of [D-Cys(tBu)(5)]NPS alone prolonged the anesthetic duration of ketamine and propofol. These results suggested that NPS decreased the anesthesia time through the NPS receptor. Another study examined the effect of NPS on an electroencephalogram during propofol and ketamine anesthesia [35]. In this study, NPS (1 or 2 nmol, icv) decreased the delta and slow-wave (SWS) power during propofol anesthesia. Similarly, NPS (1 or 5 nmol, icv) decreased ketamine-induced delta and slow-wave (SWS) power. In rats with physiological saline icv, propofol itself resulted in a remarkable increase in the delta (0.5–4 Hz) activity and a decrease in the theta (4.5–8.5 Hz) and high frequency (14.5–60 Hz) activities, whereas in rats with NPS 1 nmol icv, the duration of dominant delta activity was decreased. On the other hand, rats who received ketamine showed increased delta and theta activities and decreased high frequency (14.5–60 Hz) activity. Neuropeptide S at 1 nmol icv inhibited the ketamine-induced increase in delta power. It also enhanced the activation of the theta activities and did not affect the anesthetic-induced changes in the number of SWS; however, NPS shortened the SWS duration. The shortening characterized the depression effect of NPS for the anesthetic-induced SWS for the SWS episode period. Administration of [D-Val(5)] NPS (20 nmol) significantly antagonized the wake-promoting effect of 1 nmol NPS, but [D-Val(5)] NPS alone did not influence SWS. These results indicate that the inhibitory effect of NPS on the anesthesia duration and EEG were through NPSR [19]. As the contracting effect of NPS on anesthesia was antagonized by a co-administered NPSR antagonist, the NPS effect is considered to develop through the NPS receptor. 

### 2.4. Induction and Emergence from Anesthesia Are Not a Simple Mirror

The effect of NPS on the duration of anesthesia is similar to orexin. They both affect endogenous sleep–wakefulness circuits and facilitate anesthetic emergence. This NPS effect was observed with various anesthetics. In other words, the sedative mechanism of the anesthesia could have a common mechanism. The endogenous sleep circuit is one of the candidates for this mechanism. Propofol and ketamine affected the content of the brain orexin and melamine-concentrating hormone, an endogenous sleep-promoting substance [36]. This result provides further evidence that there could be a link between sleep and anesthesia. However, we should be aware that the endogenous sleep circuit and other mechanisms could be responsible for the anesthesia-induced loss of consciousness because NPS or orexin does not completely inhibit the sedative effect of the anesthesia. In addition, both NPS and orexin facilitate emergence from anesthesia but do not influence the time lag from anesthetic administration to the loss of the lighting reflex (introduction time) [30,33,37]. These results suggest that induction and emergence from anesthesia are not a simple mirror of the same mechanism. NPS is a useful tool in studying the mechanism of anesthesia-induced loss of consciousness.

### 2.5. Perspective: NPS in Anesthesia

General anesthesia is not uniform; therefore, the mechanism of general anesthesia is diverse. However, many reports show that activation of the endogenous sleep-promoting circuit is included in the mechanism of general anesthesia-induced loss of consciousness. It was revealed that NPS modifies endogenous sleep-promoting circuits and can facilitate anesthesia emergence (see NPS in sleep). Neuropeptide S influenced general anesthesia, promoting emergence from different types of anesthetic agents, including thiopental and ketamine [37].

Interestingly, NPS did not influence the induction time of these anesthetics at all. These results were similar to the effects of OX, another endogenous wake-promoting substance. Orexin did not affect the induction time of various anesthetics such as thiopental, ketamine, or propofol, but did promote emergence from anesthesia [30,31,33,34]. As seen from these results, the processes of induction and the emergence from anesthesia are not mirrored.

Furthermore, we report that neither the OX [30] nor NPS antagonists [37] influenced the induction time of thiopental and ketamine. These studies indicate that the endogenous sleep-promoting circuit is involved in the recovery process of anesthesia-induced loss of consciousness. As for the role of endogenous sleep-promoting substances in the mechanism of anesthesia, the details are not yet known. Parallel studies of the roles of sleep-promoting and wake-promoting substances on the mechanism of anesthesia could advance knowledge in this field. Candidate endogenous sleep-promoting substances will include melanin-concentrating hormone [36] and prostaglandin D2 [38].

The application of NPS to experimental medicine and clinical anesthesia will be significant in the future. Neuropeptide S promotes wakefulness, as already described. Thus, NPS itself and its derivatives may be helpful for developing drugs that support smooth, gentle recovery from anesthesia.

No complete antagonist to the anesthetic agent is available because the detailed mechanism of anesthesia-induced loss of consciousness is still not elucidated. It is possible that NPS may not be used as an antagonistic agent to anesthetics. However, the effects of NPS, namely, promotion of wakefulness, anti-anxiety effects, and analgesia, are helpful in smooth, gentle recovery from general anesthesia. We believe that the study of NPS in anesthesia emergence is worth continuing in the future.

## 3. NPS in Analgesia

Rats who received exogenous NPS by icv showed analgesic effects [39]. The nociceptive effect in the first period and the second phase was decreased in a dose-dependent manner with NPS icv (0.1–100 pmol). The analgesic effect of NPS (10 pmol, icv) was antagonized by co-administered [D-Val(5)]NPS (1000 or 10,000 pmol), an NPS receptor antagonist; however, [D-Val(5)]NPS alone had no hyperalgesic nor analgesic effect. Naloxone (10 mg/kg, intraperitoneal injection) did not inhibit the NPS analgesic effect at all. Neuropeptide S (10 pmol, icv) facilitated c-Fos expression in the periaqueductal gray (PAG) of formalin-administered mice, compared with saline-administered control mice. These results indicate that NPS exhibited an analgesic effect on formalin-induced nociception via NPSR, and PAG activation was responsible for this analgesic action. The NPS–NPSR system is a potential target for new pain killers [40]. Anti-nociceptive neural mechanisms are diverse; for example, opioids act at the spinal cord level [41]. As naloxone did not affect the NPS analgesic action on formalin-induced nociception in mice, the NPS–NPSR system appeared to have little interaction with the opioid anti-nociceptive system. Therefore, we question what system is involved in the sedative mechanism of NPS. 

### 3.1. Noradrenergic Neuronal Activity in the LC Interacts with NPS in Analgesia 

Many analgesics act on the descending inhibitory system [42]. The descending system consists of several nervous systems, including the noradrenergic [43] and serotonergic [44] nervous systems. A previous study suggested that the descending noradrenergic nervous system is included in one of the analgesic mechanisms of NPS. *N*-(2-Chloroethyl)-*N*-ethyl-2-bromobenzylamine (DSP-4), a selective LC noradrenergic neurotoxin, produced neuropathy. Rats pretreated with DSP-4 were evaluated to determine the analgesic effect of NPS using hot-plate and tail-flick studies [45]. Intraperitoneal injection of DSP-4 at 50 mg/kg decreased hot-plate latency but not tail-flick latency. Injection of NPS alone (0.0, 1.0, 3.3, and 10.0 nmol, icv) prolonged hot-plate latency in a dose-dependent manner. Neuropeptide S (10 nmol, icv) prolonged the hot-plate (%MPE) latency at 30 and 40 min after NPS administration; however, NPS at any dose did not affect the tail-flick latency. The analgesic effect of NPS in the hot-plate study, but not the tail-flick latency study, was antagonized by co-administered DSP-4. The DSP-4 dose decreased the quantity of noradrenaline in the cerebral cortex, pons, and hypothalamus. There was a significant relationship between the cerebrocortical noradrenaline content and hot-plate latency (a percentage of the maximal effect: %MPE) (*p* = 0.017, r2 = 0.346) [28]. The effect of DSP-4 on tail-flick latency was not changed with or without NPS in this study. The hot-plate latency reflects a supratentorial analgesic mechanism, whereas the tail-flick latency is thought to indicate the analgesic mechanism at the spinal cord level [46]. The cerebrocortical noradrenaline nerve is purely innervated from the LC. Based on these results, NPS produced an analgesic mechanism interacting with the noradrenergic neuronal activity that originated from the LC, including the descending inhibitory system [45]. 

### 3.2. Dopaminergic Neurons in NPS Analgesia

In addition to the descending noradrenergic neurons, the dopaminergic neurons are involved in the analgesic mechanism of NPS. A study investigated the role of dopamine receptor signal transduction in the anti-nociceptive property of NPS using the formalin test in mice. Several types of dopamine receptor blockers were used to determine whether dopaminergic neuronal activity is involved in the anti-nociceptive effect of NPS. Supra-spinal NPS (0.1 nmol, icv) administration significantly decreased formalin-induced nociception in the first and second phases. Morphine (5 mg/kg, sc) and indomethacin (10 mg/kg, ip) were used for a positive control. Morphine showed an anti-nociceptive effect in the first and second phases, while indomethacin showed an anti-nociceptive effect only in the second phase. Systemic administration of SCH 23390 (selective dopamine D1 antagonist, 0.05 mg/kg, ip) slightly inhibited the effect of NPS in the second phase. The systemic administration of haloperidol (nonselective dopamine D2 receptor antagonist, 0.03 mg/kg, ip) inhibited NPS-induced anti-nociceptive action. Like haloperidol, systemic sulpiride (selective dopamine D2 receptor antagonist, 25 mg/kg, ip) also denied the anti-nociceptive effect of NPS in both phases of the formalin test. These results suggest that the analgesic effect of NPS was associated with dopamine neuronal transmission through the dopamine D2 receptor signal pathway [47]. Persistent pain causes anxiety-like behavior and nervous activity; however, the detailed mechanism of anxiety-like behavior which emerges due to persistent chronic pain is still unknown. Rats in a chronic pain model of peripheral neuropathy decreased in intracerebral NPS release. If the rat was given NPS icv, pain and anxiety-like actions were alleviated. 

### 3.3. Other Neuromodulators in NPS Analgesia

Neuropeptide S promotes GABA release from interneuron to amygdala at the cellular level; therefore, changes in the GABAergic neuronal activity due to NPS resulted in a change in anxiety-like behavior [48]. Electric acupuncture (EA) showed a beneficial effect on chronic inflammatory pain and pain-related anxiety. Neuropeptide S may be included in this EA analgesic and anxiolytic effect. In a rodent inflammatory pain model, inflammation decreased the paw withdrawal thresholds (PWTs) and developed anxiety-like behavior. Ipsilateral NPS and NPSR were decreased in the anterior cingulate cortex (ACC), involved in emotional control. Conversely, EA stimulation increased the paw withdrawal thresholds (PWTs) and prevented anxiety-like behavior. Administration of NPS had a similar effect to EA on the pain model. The effect of NPS on PWTs and anxiety-like behavior was antagonized with co-administered NPSR inhibitor. The results indicate that EA improved inflammatory pain and anxiety-like behavior related to pain through certain interactions with the brain NPS–NPSR system. [49]. In addition to inflammatory pain, NPS could affect other types of pain. The rat spinal nerve ligation (SNL) model is a neuropathic pain model. In this model, the NPS of the amygdala and the NPS immunity positive cell density are decreased. A study indicated that these decreases could be involved in a pain development mechanism of SNL. In the SNL rats, NPS (1, 10, and 100 pmol/side) or saline was administered through a cannula into the amygdala at 3, 6, 9, 12, 15, and 18 days after ligation. Administration of NPS increased the thermal withdrawal latency (TWL) and mechanical withdrawal threshold (MWT) from 11 to 21 days after ligation. These results suggest that NPS at the amygdala produced anti-nociceptive and anxiolytic action. This effect of NPS was antagonized by co-administered SHA 68, a non-peptide NPS receptor antagonist. The SNL procedure facilitated expression of inflammatory mediators, NFκB p65, and CX3C chemokine receptor 1 in the amygdala. Neuropeptide S significantly inhibited this expression. The effect of NPS was antagonized by SHA 68. Micro-infusion of NPS into the amygdala attenuated symptoms of neuropathic pain, anxiety-like behavior, and may inhibit spinal cord microglia after spinal nerve injury and astrocytic response [50]. This study provided a new aspect of NPS application in pain medicine. Recently, one report showed that NPS initiates a sequential cascade reaction among orexin, substance P, glutamate, and endocannabinoids. Glutamate then activates perisynaptic mGlu5Rs to initiate the endocannabinoid retrograde inhibition of GABAergic transmission in the periaqueductal gray, leading to stress-induced analgesia in restrained mice [51]. This result suggests NPS may orchestrate the actions of various nociceptive- and antinociceptive-related substances.

### 3.4. Perspective: NPS in Pain

The development of analgesics without side effects is an earnest desire of the human being. Current clinically available analgesics are classified as opioids, steroids, and non-steroidal anti-inflammatory drugs (NSAIDs).

Among these agents, NSAIDs are a widely used pain killer with strong effect and low potential for abuse. However, NSAIDs have undesirable side effects such as liver dysfunction, renal dysfunction, and gastrointestinal ulceration. One case may have to cancel the use of NSAIDs because of these side effects. In addition, steroids have potent anti-inflammatory action. Inflammation has four symptoms: dolor, calor, tumor, and rubor. The analgesic effect of steroids is based on the inhibition of these four symptoms. However, steroids also have undesirable and severe side effects. These side effects include, for example, infection, impaired glucose tolerance, bleeding of the digestive tract, and central obesity. In some cases the use of steroids had to be cancelled due to these side effects, similar to the NSAID case above.

In addition, the long-term use of steroids results in adrenal cortical insufficiency and sometimes becomes fatal. Steroids cannot be used unconditionally as an analgesic in all cases. Opioids remain superior among the currently available analgesics. Opioids were a representative analgesic for a long time before steroids and NSAIDs were available for clinical use. However, opioids also have serious adverse effects. These effects include drowsiness, respiratory depression, constipation, and drug dependence. Above all, respiratory depression is fatal. Therefore, it is crucial to synthesize analgesics that do not cause respiratory depression. Potential for abuse must be considered as well. Thus, medical personnel hesitate to use opioids; therefore, some cases would have a lower quality of life. Neuropeptide S has both wakefulness-promoting and analgesic effects. If NPS-containing analgesics with no respiratory depression are developed, they may provide significant benefit to humans. Additionally, NPS-containing analgesics may relieve somnolence, another side effect of the opioid. Orexin prevented the somnogenic effects of morphine without any inhibition of the analgesic effect [52]. Thus, we may also apply NPS or its derivatives as adjuvants of opioids.

## 4. NPS in Sleep

### 4.1. Overview of Sleep

Although we spend approximately one-third of our whole life sleeping, it is still difficult to define sleep [53,54]. Normal sleep is classified into two different states, rapid eye-movement (REM) sleep and non-REM (NREM) sleep. Aserinsky and Kleitman discovered REM sleep in the early 1950s [55,56]. Rapid eye-movement sleep is characterized by EEG desynchronization, muscle atonia, and periodic rapid eye movements. Vivid dreaming also develops during REM sleep. Conversely, NREM sleep is characterized by high-amplitude, low-frequency EEG pattern (synchronous EEG), low skeletal muscle tone, and minimum psychological activity. The synchronous EEG includes sleep spindles, slow-wave, and k-complexes [57]. In humans, NREM sleep is divided into four subgroups (stage 1, 2, 3, and 4) based on the EEG pattern [58]. Stage 4 NREM sleep is the deepest sleep. Sleep usually begins from light NREM sleep, then progresses through deeper NREM sleep and REM sleep. This NREM–REM sleep cycle is commonly observed among mammals. The sleep cycle in adult humans is approximately 90 min in length. Rapid eye-movement sleep usually develops after NREM sleep. Sleep is a reversible behavioral state lacking response to environmental stimuli. Several brain regions are responsible for sleep processes. This list includes the thalamus, the hypothalamus, the brainstem, the basal forebrain, and the cerebral cortex [59]. Traditionally, Morruzzi and Magoun indicated that an ascending arousal system in the brainstem is responsible for wakefulness. They showed sensory inputs activated the brain and consciousness [60]. A recent hypothesis suggests that the sleep-promoting and wake-promoting systems collaborate to control sleep–wakefulness [61]. The preoptic area of the hypothalamus inhibits wake-promoting systems arising from the upper portion of the brainstem. In contrast, the ascending arousal system can also inhibit the preoptic area’s sleep-promoting system. These two systems act as a ‘flip-flop’ switch that can rapidly change consciousness between sleep and wakefulness [62]. Several brain regions are involved in endogenous sleep regulatory neural pathways [61]. The regions include the basal forebrain (BF), the lateral hypothalamic area (LH), the locus coeruleus (LC), the pedunculopontine tegmental nucleus (PPTg), the perifornical area (Pef), the ventrolateral POA (VLPO), the dorsal raphe nucleus (DRN), and the tuberomammillary nucleus (TMN). The TMN and VLPO are key components of the ‘flip-flop’ switch. Neuronal firing activity of the TMN is higher during wakefulness and lower during sleep. In contrast, the VLPO is active during sleep and relatively silent during wakefulness. Both nuclei reciprocally act to induce sleep and wakefulness, similar to a ‘flip-flop’ switch of sleep and wakefulness. Prior to sleep onset, sleep-active neurons from the VLPO release GABA in the various regions described above to inhibit their activity. Conversely, neurons from the TMN release histamine in the cerebral cortical and subcortical structures to initiate wakefulness. Many neurotransmitters are involved in these systems. For example, acetylcholine in the PPTg, noradrenaline in the LC, serotonin in the DRN, dopamine adjacent to the DRN, and histamine in the TMN are involved [61,62,63].

### 4.2. NPS and NPSR Roles in Sleep Regulation

NPS and the NPS receptor are included in sleep regulation. A report showed that transmutation of the NPS receptor gene caused primary insomnia in humans. Single-nucleotide polymorphism (SNP) variations resulted in sleep changes in humans [64]. Polymorphisms of the NPSR1 gene might cause susceptibility to primary insomnia. At present, two different allelic combinations (CATGTC and GCCAAT) were identified as a causative factor of this primary insomnia [9]. Genetic mutations of the NPS receptor inhibited sleep in rodents; however, the sleep requirement persisted. These animals presented no memory impairment associated with sleep disorders [10]. These results indicate that the wake-promoting effect of NPS is consistent across species. The role of NPS in memory retention and sleep regulation was not similarly consistent. Neuropeptide S promotes wakefulness at the ventrolateral preoptic area (VLPO), one of the sites associated with non-REM sleep regulation. Neuropeptide S perfusion into the anterior hypothalamus, including the VLPO, promotes wakefulness. The blood vessels were reversibly contracted with NPS perfusion in brain slices, including the VLPO; therefore, NPS reversely inhibits VLPO activity. The NPS receptor is present in the GABAergic neuron of the VLPO. Neuropeptide S facilitates the GABAergic neurons in the VLPO. The GABAergic neurons act as inhibitory neurons in sleep regulation; thus, NPS acts as an indirect inhibitor of sleep-promoting neurons. As a result, NPS acts as a wake-promoting substance [65]. Another study showed that NPS (0.1 and 1 nmol, icv) significantly increased wakefulness at the first two hours post NPS injection. This wakefulness is characterized by increased EEG high-frequency activities (14.5–60 Hz). NPS also enhanced c-Fos expression, as compared with saline injection control, in the posterior hypothalamic histaminergic neurons by 76.0%, in orexinergic neurons by 28.2%, in the perifornical nucleus (PeF) by 24.3%, in the dorsomedial hypothalamic nucleus (DMH) by 13.7% in the lateral hypothalamic area (LH) of the posterior hypothalamus [66]. The NPS-induced c-Fos expression in histaminergic neurons and orexinergic neurons where the NPS receptor is highly expressed suggests that NPS activates histaminergic and orexinergic neurons to promote wakefulness. These results suggest that NPS could play a role as a conductor which orchestrates these sleep–wakefulness-related brain structure–activity relationships to promote quantified sleep rhythm (Figure 1). Neuropeptide S inhibits sleep and promotes wakefulness. As mentioned above, there are two distinct sleep statuses: rapid eye movement (REM) sleep and non-rapid eye movement (NREM) sleep. The REM sleep-regulation mechanism is different from that of NREM sleep. Non-rapid eye movement sleep emerges from the cerebral cortex and thalamus complex [59]. On the other hand, the lower brain portion, including the brainstem, controls REM sleep [67]. Therefore, there is a difference between a study on the role of NPS in REM or NREM sleep. The detailed roles of endogenous NPS in these two different sleep statuses are still unknown. One study showed that NPS affects these sleep statuses differently. A study tested the effect of the NPS receptor antagonist [D-Cys((t)Bu)(5)]NPS (2 and 20 nmol, icv) on physiological sleep and spontaneous locomotor behavior. The higher dose of [D-Cys((t)Bu)(5)]NPS decreased the amount of time spent in wakefulness and increased the time spent in NREM sleep. There was no statistically significant difference on the time spent in REM sleep. The antagonist produced no behavioral changes, including abnormal gross motor behavior [68]. The results indicated that endogenous NPS regulates physiological sleep and that its roles in NREM sleep and REM sleep are different. Typically, REM sleep will develop after deep enough NREM sleep. In this study, NPS inhibited NREM sleep without any effect on REM sleep; therefore, NPS might inhibit NREM sleep during the light phase but not during the dark phase. Further study is required to clarify this theory. Neuropeptide S and its antagonist can be useful tools in studying physiological sleep.

### 4.3. NPS and NPSR Roles in Sleep Disturbance

The roles of NPS in behavior can be separated from its roles in sleep. Animals with sleep disorders typically exhibit anxiety-like behavior. The amygdala is thought to be responsible for the development of anxiety-like behavior. According to Xie et al., the amygdala is involved in the anxiety-like action of sleep disorders [69]. The inhibition of REM sleep for 24 h induces anxiety-like behavior. The subsequent sleep rebound increases the EEG theta band frequency. This rebound results in an increase in the amount of NREM sleep during the light phase and results in an increase in the length and amount of NREM sleep in the dark phase. Neuropeptide S at 1 nmol icv decreased the sleep disorder-induced anxiety-like action and inhibited electroencephalographic theta band frequency. The administration of NPS also inhibited REM sleep and promoted wakefulness. The inhibition of REM sleep upregulated the expression of mRNA in the NPS receptor in the amygdala. This mRNA enhancement is found in the basolateral amygdala, central amygdala, and the medial amygdala. Such upregulation of the NPS receptor mRNA could induce a counteraction of NPS to sleep disturbance-induced anxiety-like behavior [69]. Neuropeptide S regulates sleep and anxiety. Each neuron has its own function; therefore, the excitement degree of an individual NPS neuron is proportional to the activity of the region in which NPS is innervated. For example, the NPS neuron in the locus coeruleus is involved in REM sleep regulation. Indeed, NPS transcript levels were increased only in the peri-coerulear group in sleep-deprived rats who were tested with the single platform-on-water (mainly REM) sleep deprivation method, but not in stress controls [70]. Neuropeptide S improved sleep-deprivation-induced memory impairment. One study examined whether 1 nmol NPS improved memory impairment with twenty hours of sleep deprivation. A mouse was examined using T-maze learning with or without NPS (1 nmol, icv). The sleep restriction reduced the neuronal activity of the mouse cingular cortex and inhibited the process of short-term memory. One nmol of NPS counteracted sleep deprivation-induced memory impairment by activating the neuronal activity of the cingular cortex responsible for memory-related processes [71]. This sleep disorder resulted in several sequelae across various systems, including the respiratory, cardiovascular, and central nervous systems. For example, bronchial asthma, ischemic heart disease, and cerebral stroke attack were developed during the post-operative period [72]. Sleep disturbance developed into post-operative cognitive disturbance (POCD). A patient with POCD also had a poor prognosis; therefore, treatment of POCD is a key point to improve in the post-operative course. 

### 4.4. NPS–NPSR System: A Possible Research Target to Improve Surgical Stress-Induced Sleep Disturbances 

It is not clear whether surgical stress itself affects sleep architecture because the application of any surgical procedure without any type of anesthesia is not ethically acceptable; however, sleep disturbances have developed from surgery with local anesthesia [73]. Thus, surgical stress alone could result in sleep disturbances. As NPS has an anti-stress action [74], the NPS–NPSR system is a research target to improve surgical stress-induced sleep disturbances. Opioids are widely used as common analgesics for surgical wounds because opioids have a prominent analgesic effect. On the other hand, morphine disturbs REM sleep [75]; thus, the use of morphine aggravates the post-operative course. Physicians may avoid administering morphine to a patient because of respiratory depression and over-sedation [76]. A recent study indicated that endogenous wake-promoting substances may prevent the adverse effects of opioids, as mentioned above. Indeed, orexin, an endogenous wake-promoting peptide, inhibited over-sedation by morphine in rats, while orexin had no interaction with the analgesic effect of morphine [52]. Similar to orexin, NPS promotes wakefulness and analgesia without any respiratory depression. We could expect NPS to establish a method for opioid administration with no adverse effects. A study showed that NPS improved morphine withdrawal symptoms [77]. This result indicates that NPS interacts with the action of opioids. Although the details of this interaction remain to be clarified, the NPS role in opioid action is a promising field in NPS research. In addition to opioid or surgical stress, general anesthesia results in sleep disturbances. However, the property of general anesthetics is not uniform but heterogeneous. One anesthetic may act as a GABA receptor agonist, and another may act as an antagonist to the NMDA receptor. Others may act on various sites [24]. Therefore, different types of anesthetics produce different changes in the sleep architecture. We previously reported that propofol, a GABAergic anesthetic, inhibited wakefulness in rats, while ketamine, an NMDA antagonist, enhanced wakefulness and inhibited non-rapid eye movement sleep (NREMS) during that period. Ketamine decreased the orexin content in the hypothalamus during the anesthesia period. The orexin content was restored to pre-anesthesia levels in the hypothalamus and pons. Both propofol and ketamine increased the brain melanin-concentrating hormone content, an endogenous sleep-promoting substance in the postanesthetic period, with the degree of increase being greater with propofol [36]. As NPS has a similar property in the sleep–wake rhythm as orexin does, anesthetics and NPS could affect each other. Studies regarding the role of NPS in post-anesthetic sleep disturbances could contribute to developing a better post-operative course.

### 4.5. Perspective: NPS in Sleep

NPS has various physiological functions. Wakefulness-promotion is one of the main effects among them. Numerous brain structures, neurotransmitters, and neuromodulators cooperate to form the sleep–wakefulness cycle, as explained in the overview of sleep. Neuropeptide S orchestrates them. Sleep disorders result in ischemic heart disease, cerebrovascular disorder, disorders of carbohydrate metabolism, and cognitive impairment [78]. These result in a health hazard. Additionally, the health hazard of the shift system worker is serious. The hypnotic agent synthesized from endogenous substances is expected to have few side effects. In fact, some OX receptor antagonists have already been applied as hypnotics [79]. We may also develop the derivatives of NPS as risk-free hypnotics without side effects. Sleep disorders occur after anesthesia [78] and surgery [72]. Also, opioids are known to cause sleep disorders [80,81]. Opioids are an indispensable analgesic during the post-operative period but cause sleep disorders that worsen the post-operative course. We may relieve these sleep disorders by applying NPS. There is a possibility for the use of NPS in the treatment of normal sleep disorders and post-operative sleep disorders. For future studies of NPS as sleeping drugs may be of value.

## 5. Conclusions

We reviewed the roles of NPS in three domains: anesthesia, analgesia, and sleep regulation. Neuropeptide S has various and diverse physiological properties, and plays a crucial role in these three domains (Figure 2). We expect that further study of NPS will contribute to the development of basic medicine and will lead to better medical care.

## Figures and Tables

**Figure 1 pharmaceuticals-14-00483-f001:**
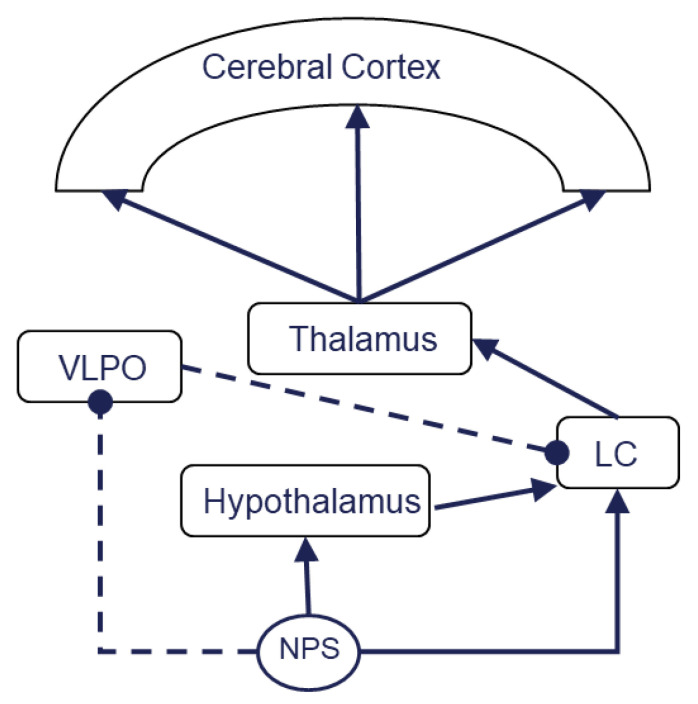
NPS orchestrates an endogenous sleep–wakefulness circuit. Activation of locus coeruleus results in excitation of the thalamus, then cerebral cortex activation. This sequential process facilitates wakefulness. VLPO inhibits wakefulness by LC depression. NPS activates the hypothalamus and LC but inhibits VLPO. NPS could play a role as a conductor which orchestrates these sleep–wakefulness-related brain structure–activity relationships to promote quantified sleep rhythm. The solid line indicates an exciting process, while the dotted line denotes inhibition; LC, the locus coeruleus; VLPO, the ventrolateral preoptic area.

**Figure 2 pharmaceuticals-14-00483-f002:**
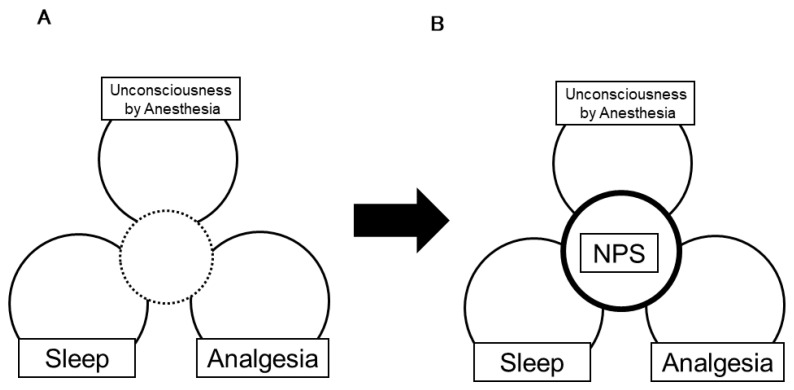
(**A**) Conventionally, there is a weak link among anesthesia-induced unconsciousness, physiological sleep, and analgesia. (**B**) The physiological functions of endogenous peptides in the brain, including NPS, suggest that they share common mechanisms. A study of the role of NPS in these three domains will help to determine the details of how these three domains interact with each other.

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
