# Peer review of "Roles of Neuropeptide S in Anesthesia, Analgesia, and Sleep"

_pharmaceuticals, 2021, doi:10.3390/ph14050483_

Round 1

Reviewer 1 Report

Neuropeptide S (NPS) is a selective 20-amino acid ligand for the NPS receptor (NPSR), a Gs and Gq – coupled receptor formerly identified as orphan receptor GPR 154. NPS is an endogenous modulator of a wide spectrum of physiological activities in the brain. It exerts highly anxiolytic effects, stabilizes arousal state, regulates food intake as well as plays a role in the pathomechanism of fear modulation and addiction. This interesting and review paper by Kushikata and colleagues  comments the role of NPS in the central mechanisms of sleep, anaesthesia and analgesia. Presented work seems to be valuable for all working in the field of neurendocrinology and pharmacology.  It may potentially help to introduce some novel therapeutical strategies in the treatment of sleed disorders and open new possibilities in contemporary anaethesiology. An outline characteristics of NPS related   signalling is complete and very informative for the readers. To sum up, this review can be classified as important and original contribution to the field of neuroscience.

However, I have  just one suggestions for the Authors:

Chapter „NPS in sleep”. This part of the review would benefit from adding a scheme depicting an involvment NPS in the central sleep-wake regulatory pathway (at the level of hypothalamus and brainstem).

Author Response

Author's Reply to the Review Report (Reviewer 1)

Neuropeptide S (NPS) is a selective 20-amino acid ligand for the NPS receptor (NPSR), a Gs and Gq – coupled receptor formerly identified as orphan receptor GPR 154. NPS is an endogenous modulator of a wide spectrum of physiological activities in the brain. It exerts highly anxiolytic effects, stabilizes arousal state, regulates food intake as well as plays a role in the pathomechanism of fear modulation and addiction. This interesting and review paper by Kushikata and colleagues comments the role of NPS in the central mechanisms of sleep, anaesthesia and analgesia. Presented work seems to be valuable for all working in the field of neurendocrinology and pharmacology.  It may potentially help to introduce some novel therapeutical strategies in the treatment of sleed disorders and open new possibilities in contemporary anaethesiology. An outline characteristics of NPS related   signalling is complete and very informative for the readers. To sum up, this review can be classified as important and original contribution to the field of neuroscience.

However, I have  just one suggestions for the Authors:

 Chapter „NPS in sleep”. This part of the review would benefit from adding a scheme depicting an involvment NPS in the central sleep-wake regulatory pathway (at the level of hypothalamus and brainstem).

We made a scheme showing how NPS plays in sleep-wakefulness in CNS, focusing on the hypothalamus and brainstem. We hope this scheme could provide more helpful information on the role of NPS easily to the readers of pharmaceuticals.

Reviewer 2 Report

This is a well planned and executed review. The writing is clear, however, some of the paragraphs are very long. Since this is primarily style choice, I will not address it here, however.

In the introduction, however, there needs a little more reference to receptor classification.

Where you state that:

'NPS receptor 1 (NPSR1) is expressed primarily in the 22
bronchus, brain, and immune cells.'

Better to say:

NPS receptor 1 (NPSR1) is a G protein-coupled receptor (GPCR) that is expressed primarily in the 22 bronchus, brain, and immune cells (X).

Similarly, later in the paragraph, there needs to be a citation:

For example, a single-nu-cleotide polymorphism of NPSR1 is significantly associated with disease, such as asthma, anxiety and emergence disorder, and rheumatoid arthritis (X).

Cite:

(X) Thompson, M.D., Hendy, G.N., Percy, M.E., Bichet, D.G., Cole, D.E.C. G protein-coupled receptor mutations and human genetic disease (2014) Methods in Molecular Biology, 1175, pp. 153-187.

Author Response

Author's Reply to the Review Report (Reviewer 2)

This is a well planned and executed review. The writing is clear, however, some of the paragraphs are very long. Since this is primarily style choice, I will not address it here, however.

In the introduction, however, there needs a little more reference to receptor classification.

Where you state that:

NPS receptor 1 (NPSR1) is expressed primarily in the bronchus, brain, and immune cells.

Better to say:

NPS receptor 1 (NPSR1) is a G protein-coupled receptor (GPCR) that is expressed primarily in the bronchus, brain, and immune cells (X).

We revised the sentence as the reviewer pointed out. We tried to put subhead at each paragraph and divide one long paragraph into a few one. We thank for the comment.

Similarly, later in the paragraph, there needs to be a citation:

For example, a single-nucleotide polymorphism of NPSR1 is significantly associated with disease, such as asthma, anxiety and emergence disorder, and rheumatoid arthritis (X).

Cite:

(X) Thompson, M.D., Hendy, G.N., Percy, M.E., Bichet, D.G., Cole, D.E.C. G protein-coupled receptor mutations and human genetic disease (2014) Methods in Molecular Biology, 1175, pp. 153-187.

We cited several references accompanied with each item. We thank for the comment.